# Reviewing Publicly Available Reports on Child Health Disparities in Indigenous and Remote Communities of Australia

**DOI:** 10.3390/ijerph20115959

**Published:** 2023-05-25

**Authors:** Kedir Y. Ahmed, Julaine Allan, Hazel Dalton, Adrian Sleigh, Sam-ang Seubsman, Allen G. Ross

**Affiliations:** 1Rural Health Research Institute, Charles Sturt University, Orange, NSW 2800, Australia; 2National Centre for Epidemiology and Population Health, Research School of Population Health, Australian National University, Canberra, ACT 2601, Australia; 3School of Human Ecology, Sukhothai Thammathirat Open University, Nonthaburi 11120, Thailand

**Keywords:** health disparities, indigenous, remoteness, children, Australia

## Abstract

Developing programs that ensure a safe start to life for Indigenous children can lead to better health outcomes. To create effective strategies, governments must have accurate and up-to-date information. Accordingly, we reviewed the health disparities of Australian children in Indigenous and remote communities using publicly available reports. A thorough search was performed on Australian government and other organisational websites (including the Australian Bureau of Statistics [ABS] and the Australian Institute of Health and Welfare [AIHW]), electronic databases [MEDLINE] and grey literature sites for articles, documents and project reports related to Indigenous child health outcomes. The study showed Indigenous dwellings had higher rates of crowding when compared to non-Indigenous dwellings. Smoking during pregnancy, teenage motherhood, low birth weight and infant and child mortality were higher among Indigenous and remote communities. Childhood obesity (including central obesity) and inadequate fruit consumption rates were also higher in Indigenous children, but Indigenous children from remote and very remote areas had a lower rate of obesity. Indigenous children performed better in physical activity compared to non-Indigenous children. No difference was observed in vegetable consumption rates, substance-use disorders or mental health conditions between Indigenous and non-Indigenous children. Future interventions for Indigenous children should focus on modifiable risk factors, including unhealthy housing, perinatal adverse health outcomes, childhood obesity, poor dietary intake, physical inactivity and sedentary behaviours.

## 1. Introduction

Health disparities refer to differences in health status among different groups of people that stem from inequalities in their living conditions, such as where they were born, grew up, live, work and age [1,2]. These inequalities can result from various factors such as social, economic, environmental and structural disadvantages related to income, ethnicity, education, age, gender, employment, disability and migration status [3]. The root causes of health disparities are complicated and in flux, but they are interrelated and related to socioeconomic status [2]. Health inequities are strongly associated with poor access to quality medical care, limited personal and social support systems, higher rates of addiction, family violence and poor housing conditions [2,4,5]. The impacts of health disparities are not confined to the individual level, but also have effects on a nation’s economic performance. For example, an economic analysis from the United States of America (USA) indicated that health disparities cost approximately $42 billion in lowered productivity and $93 billion in excess medical costs each year [6].

Colonisation and racism continue to impact every aspect of Indigenous lives [7,8,9]. In Australia, Aboriginal and Torres Strait Islander peoples (hereafter, we use “Indigenous” as a collective term, respectfully acknowledging the diversity of language and culture of Aboriginal and Torres Strait Islander peoples, as the First People and custodians of Australia) have faced ongoing challenges resulting in a number of health disparities relative to the non-Indigenous populations [10]. Chronic health conditions (including diabetes, arthritis, obesity and vascular diseases) are higher in Indigenous populations compared to non-Indigenous populations [11,12,13,14,15]. Indigenous Australians also experience greater impacts from risk factors such as smoking, risky alcohol consumption, limited consumption of fruits and vegetables and physical inactivity [16,17,18,19]. A higher level of environmental and socioeconomic disadvantages (including poor housing, lower incomes, higher unemployment and exposure to droughts and floods) are experienced by Indigenous Australians [20,21,22].

In recognising these facts, in 2008, the first ‘Closing the Gap’ agreement was approved by the Council of Australian Governments (COAG) with six targets, including two health-related outcomes: to close the life expectancy gap within a generation and to halve the gap in mortality rates for Indigenous children under five within a decade [23]. In 2020, the Closing the Gap commitment was revised and signed by the National Federal Reform Council and the National Coalition of Peaks Indigenous Organisation with the main aim “to overcome entrenched inequality faced by too many Indigenous Australians, so that their life outcomes are equal to all Australians” (Page 3) [24]. Despite the Australian Government’s intentions, limited improvements have been seen in Indigenous health outcomes. For example, Indigenous child mortality rates decreased by 7% between 2008 and 2018, but the gap widened due to the faster reduction in child mortality rates in non-Indigenous children [25]. The gaps in life expectancy rates for Indigenous populations also persist, with 8.6 fewer years of life for Indigenous men compared to non-Indigenous men and 7.8 fewer years for Indigenous women compared to non-Indigenous women born in 2015–2017 [25].

Programs and strategies for Indigenous children that facilitate a safer start for early life would likely give the greatest chance of success and strong foundations for better future health outcomes in Indigenous populations. To inform these programs and strategies, governments require detailed information on where to focus attention for the best results. However, in Australia, the available literature (including government reports) on health disparities has limitations. A child-specific statement on disparities (including health risk behaviours) has not been produced, and the Closing the Gap reports have only reported the overall targets of child mortality and life expectancy [25,26]. Additionally, the disparate body of information on the health of the Australian population has not been interrogated for the details of Indigenous child health [26]. A detailed understanding of health disparities between Indigenous and non-Indigenous children is essential for policy formulation and advocacy as part of the implementation of the Closing the Gap agreement and checking the progress of the National Indigenous health targets.

Accordingly, we reviewed the health disparities of Australian children in Indigenous and remote communities using publicly available reports. The specific objectives were to understand disparities related to: (i) maternal and child health outcomes; (ii) long-term health conditions and disease burdens; (iii) childhood overweight and obesity; (iv) dietary intake; and (v) physical activity and sedentary behaviours.

## 2. Materials and Methods

The review was conducted using a thorough search of the Australian government and other organisations’ websites (e.g., the Australian Bureau of Statistics [ABS], the Australian Institute of Health and Welfare [AIHW] and the World Health Organization [WHO]), electronic databases (e.g., MEDLINE/PubMed) and grey literature sites for articles, government documents and project reports that targeted Indigenous populations.

In addition to the search terms (i.e., Australia, Indigenous, Aboriginal, Torres Strait Islander, First Nations, health disparities, remote and children) that were used in the overall searching process, the following context-related terms were used: maternal and child health-related terms included smoking during pregnancy, low birth weight (LBW), teenage motherhood, teenage pregnancy, stillbirth, infant mortality, child mortality and adverse birth outcomes. Long-term health condition related terms included disease burden, mental and substance-use disorders, asthma, hearing problems, dental problems and cancer. Overweight and obesity-related terms included overweight, obesity, body mass index, obesogenic, and nutritional status. Dietary-intake-related terms included fruit consumption, vegetable consumption, sugary drinks, sweetened drinks and diet drinks. Physical activity and sedentary behaviour-related terms included physical exercise, physical activity, screen time and sedentary behaviours.

Definitions of terms used: *Crowding* was defined as households that do not meet the following requirements: no more than two persons per bedroom, children aged <5 years of different sexes may reasonably share a bedroom, children aged ≥5 years of the opposite sex should have separate bedrooms, children aged <18 years and the same sex may reasonably share a bedroom and single household members aged >18 years should have a separate bedroom, as should parents or couples [27]. *Teenage mothers* represent women who gave birth when they were aged under 20 [28]. *Adequate fruit intake* was defined as the minimum recommended number of servings of fruit per day: 1 for children aged 2–3, 1½ for children aged 4–8, and 2 for people aged 9 and over [29]. *Adequate vegetable intake* was defined as the minimum recommended number of servings of vegetables per day: 2½ for children aged 2–3, 4½ for children aged 4–8, 5 for children aged 9–11 and females aged 12 and over, and 5½ for males aged 12–18 [29]. For children aged 2–4 years, physical activity recommendations included at least 180 min a day of physical activity, including energetic play, and no more than 60 min a day engaged in screen-based activity [30]. For children aged 5–17 years, physical activity recommendations included at least 60 min a day of moderate to vigorous intensity physical activity and no more than 2 h a day of screen-based activity for entertainment (for example, television, seated electronic games and computer use) [30].

### Result Synthesis

In the present study, we used narrative descriptions of frequencies and percentages to examine and contrast the socio-demographic and health results between Indigenous and non-Indigenous communities. We also extracted 95% confidence intervals (CIs) and *p*-values from relevant sources when applicable.

## 3. Results

### 3.1. Data Sources

Our searching identified nine reports (including the 2016 and 2021 ABS censuses [31,32], a National Health Survey [33], the 2018/19 and 2012/13 National Aboriginal and Torres Strait Islander Health Surveys [34,35] and four Australian Institute of Health and Welfare Health reports [36,37,38,39]) and one original article [40]. Table 1 shows the main data sources for this study.

## 3.2. Sociodemographic and Housing Conditions

In Australia, the 2021 Census of Population and Housing showed that there are 812,728 people identified as Aboriginal and/or Torres Strait Islanders, representing 3.2% of the total population [32]. The proportion of Indigenous children aged less than 15 years was higher (32.7%) compared to the total population (18.2%) [32]. The average number of people per household was also higher in Indigenous households compared to total households (3.1 versus 2.5) [32]. The completion rate for year 12 or higher schooling was lower in Indigenous populations compared to the total population (48.7% versus 66.7%) [32] (Table 2). The 2016 Australian Bureau of Statistics (ABS) census documented that 18% of Indigenous Australians were living in crowded dwellings compared to 5% of non-Indigenous Australians [31], and nearly one in ten Indigenous households required one or more extra bedrooms in 2021 [32].

## 3.3. Maternal and Child Health Outcomes

Smoking during pregnancy was higher among Indigenous Australian mothers (43%) compared to non-Indigenous mothers (11%) [36], and in addition, the magnitude of teenage motherhood was higher among Indigenous mothers (46.4 births per 1000) compared to non-Indigenous mothers (7.1 per 1000) [36]. The rate of low birth weight (LBW) was higher among Indigenous mothers (13.0%) compared to non-Indigenous mothers (6.4%), and additionally, the infant mortality rate was higher among Indigenous infants (6.2 per 1000) compared to non-Indigenous infants (3.1 per 1000) [36]. The gap was also evident for child mortality between Indigenous and non-Indigenous children (22 versus 11 per 100,000) [36]. Developmental vulnerability in one or more domains was higher among Indigenous children compared to non-Indigenous (41% compared with 20%) [36]. Remote and very remote areas had higher rates of smoking during pregnancy, teenage motherhood, LBW, infant mortality, child mortality and developmental vulnerability compared to urban centers [36].

## 3.4. Long-Term Health Conditions

For children aged 0–14 years, a slight difference was observed in the magnitude of mental and behavioural conditions (i.e., anxiety, depression and behavioural or emotional problems) between Indigenous and non-Indigenous children (14.8% versus 11.1%) [33,34]. However, according to the 2018 Australian burden of disease study, mental and substance use disorders are the leading cause of the disease burden in Indigenous children aged 5–14 years [37]. The prevalence of asthma for Indigenous children (15%) was higher compared to non-Indigenous children (9.3%) [35], but non-Indigenous children aged 0–14 years were more likely to be diagnosed with type 1 diabetes compared to Indigenous children (137 compared with 89 cases per 100,000) [36]. Hearing impairment was reported by 6.9% of Indigenous children aged 0–14 years compared to 3.0% in non-Indigenous children [38]. Remote and very remote areas had a lower prevalence of childhood mental illness, asthma and cancer compared to major urban centers [36].

## 3.5. Overweight and/or Obesity

Overweight and/or obesity were higher in Indigenous children (37.9%) compared to all Australian children (24.9%) [33,34] (Figure 1). An Australian study also documented a higher prevalence of enlarged waist circumference (central obesity) among Indigenous children [40]. Between 2012/13 and 2018/19, overweight and obesity increased from 29.8% to 37.9% among Indigenous children [34]. Indigenous children aged 2–14 years who resided in very remote areas were less likely to be overweight or obese compared to those who were from major cities (22% compared to 36%) [36].

## 3.6. Fruit and Vegetable Consumption

The adequate daily fruit consumption rate for Indigenous children (65.0%) was lower compared to all Australian children (73.0%), and 61.2% of Indigenous children consumed two or more servings of fruit compared to 70.4% of all children [33,34]. The adequate daily vegetable consumption rate for Indigenous children (6.1%) was similar to all Australian children (6.3%) [33,34]. Nearly half of Indigenous children consumed two or more servings of vegetables (48.3%) compared to 57.3% of all children [33,34]. The daily consumption rate of sugar-sweetened drinks was 3.2 times higher in Indigenous children compared to all Australian children [33,34] (Table 3).

## 3.7. Physical Activity and Sedentary Behaviours

The proportion of children aged 5–14 who met the screen-time guidelines was almost equal for Indigenous and non-Indigenous children, but Indigenous children were more likely to meet the physical activity guidelines (54% compared with 41%, respectively) and both sets of guidelines (29% compared with 22%, respectively) [39]. Australian children living in major cities (20%) were less likely to have met the physical activity guidelines than children living in other areas (30%) [39].

## 4. Discussion

We reviewed publicly available reports on child health disparities among Indigenous and remote communities. The crowding rate was higher in Indigenous dwellings compared to non-Indigenous. Smoking during pregnancy, teenage motherhood, LBW, and infant and child mortality rates were higher among Indigenous and remote communities. Childhood obesity (including central obesity) and inadequate fruit consumption rates were higher in Indigenous children compared to non-Indigenous children. However, Indigenous children from remote and very remote areas had a lower rate of overweight or obesity, and Indigenous children also performed better in physical activity and screen-based sedentary behaviours compared to non-Indigenous children. No difference was observed in the mental and substance-use disorders and vegetable consumption rates between Indigenous and non-Indigenous children.

Poor housing has a strong correlation with child health because children are more exposed to indoor air pollution, show increased hand-to-mouth activity, and have higher respiratory rates [41,42]. Although ‘Healthy Homes’ were one of the seven action areas in the Coalition of Australian Governments’ ‘Closing the Gap’ Campaign in 2008 [43], our review showed that crowding was higher in Indigenous than non-Indigenous dwellings, as supported by previous studies in Australia [22,44]. Anderson and colleagues pointed out that unaffordable housing, homelessness, and family and kinship responsibilities are potential explanations for the higher level of crowding in Indigenous households [22]. Furthermore, our results may be explained by the lack of sufficient housing to accommodate the population size, as new houses are not being constructed at a rate that keeps up with population growth in these areas. It is imperative to address the housing crisis faced if Closing the Gap health targets are to be successfully achieved.

Parental smoking can affect the health and well-being of children through direct foetal exposure to tobacco compounds from the mother or infant exposure via breastmilk, in addition to the effects of secondhand smoke from household members [45,46]. Smoking during pregnancy has consequences including pre-term birth, LBW, miscarriage, stillbirth and anatomical anomalies [47,48]. Despite this fact, smoking during pregnancy was higher in Indigenous mothers compared to non-Indigenous mothers [36]. A narrative review by Gould et al. highlighted systemic barriers for Australian Indigenous pregnant mothers, including a lack of knowledge, confidence and skills in health providers, and failures to care for pregnant Indigenous women, as well as the lack of guidelines specific to the Indigenous context [49]. Our review warrants the need for culturally tailored antenatal educational programmes for Australian Indigenous mothers.

Teenage motherhood is associated with adverse health outcomes, including premature birth, LBW, stillbirth, neonatal death, postpartum depression and child-feeding problems [50,51]. Teenage mothers have also a higher risk of family, sexual and partner violence, depression and rapid repeat pregnancy [51]. In Australia, the magnitude of teenage motherhood was higher in remote and very remote areas and Indigenous mothers compared to major cities and non-Indigenous mothers, respectively [36]. The most common risk factors for teenage pregnancy include earlier age at first sexual intercourse [52], inconsistent or ineffective contraceptive and condom use [53], and family disruption and violence [54,55]. Our findings suggest the need for non-judgemental medical interventions and psychosocial support for teenage mothers to improve outcomes for both the young parent and their children.

Our review revealed that Indigenous Australians had a higher risk of adverse birth and child outcomes, including LBW, infant and child mortality and developmental vulnerability. Intergenerational trauma, socioeconomic disadvantages, limited medical care access and stressful life events may have also contributed to the higher level of adverse birth and child health outcomes [56]. A previously published study among Indigenous Western Australians documented that a large proportion of adverse birth outcomes were attributed to common modifiable risk factors such as smoking, alcohol or drug misuse and assault against the mother [57]. We recommend maternal Indigenous-specific programs on modifiable risk factors for adverse birth and child outcomes (including smoking, alcohol and other drugs) to reduce adverse birth and child outcomes.

Obesity is the second major contributor (next to smoking) to the health gap between Indigenous and non-Indigenous peoples in Australia [58,59]. Our study revealed that overweight and/or obesity (including central obesity) were higher in Indigenous children compared to the total of Australian children. The higher burden of overweight and obesity among Indigenous Australians could be explained by the cultural, socioeconomic and political impact of colonisation, the forced removal of Indigenous people from traditional lands and the inability to access traditional food sources [60]. Forced access to an energy-dense Western diet due to changes in the food market and lifestyle also negatively affected their dietary behaviours [60]. Obesity (including central obesity) earlier in life is associated with increased vulnerability to metabolic syndromes (type 2 diabetes, heart disease and vascular disorders), indicating the public health importance of childhood obesity [61,62].

In Australia, programmes targeting childhood obesity have been implemented, including ‘Munch and Move’ [63], ‘Live Life Well @ School’ [64], ‘Go4Fun’ [65] and ‘Parenting, Eating and Activity for Child Health’ [66]. However, most of the programmes are not tailored to Indigenous children and the effectiveness of these programmes for Indigenous children has not been determined [58]. To address the current gap in childhood obesity between Indigenous and non-Indigenous children, the government should co-design Indigenous-specific health promotion and obesity prevention programmes in schools and communities.

Inappropriate dietary intake (including inadequate fruits and vegetables, and sugary drink consumption) accounted for a large proportion of disease burdens in Indigenous Australians [37]. Our review also showed that the adequate daily fruit consumption rate was lower in Indigenous children compared to the total of Australian children [33,34]. The daily sugar-sweetened drinks consumption rate was 3.2 times higher in Indigenous children compared to the total of Australian children [33,34]. Previous studies have indicated that Indigenous peoples have socioeconomic, historical and political barriers to accessing and using healthy diets, including fruits and vegetables [67,68]. A qualitative study by Thurber et al. pointed out that children who resided in remote areas were as much as ten times more likely to face accessibility barriers, including the cost, availability and quality of fruits and vegetables [69]. In Northern NSW, a ‘fruit & Veggie program’ was implemented to improve the health and well-being of Indigenous children with poor diets. The program provides subsidised boxes of fruits and vegetables weekly for a family in need. Expanding similar programmes would be helpful to address the barriers to accessing healthy diets for Indigenous Australian children.

### Strengths and Limitations of the Study

Our review has limitations that need to be taken into consideration while interpreting the findings. Firstly, we used data from different sources (e.g., Australian National Health Survey and National Aboriginal and Torres Strait Islander Health Survey) to compare Indigenous and non-Indigenous children. Nevertheless, our findings would provide baseline information for further investigations and Indigenous-specific programmes. Secondly, teenage motherhood is different from teenage pregnancy, as teenage motherhood excluded stillbirths, miscarriages and terminations. Thirdly, high non-response rates for height and weight measurements in the national surveys limited the representativeness of the estimates. Despite the above limitations, the study provided evidence of child health disparities for policymakers and public health practitioners in addressing Indigenous children and those who reside in remote and very remote areas.

## 5. Conclusions

Our findings showed crowding rates were higher in Indigenous dwellings, and in addition, smoking during pregnancy, teenage motherhood, LBW, and infant and child mortality rates were higher among Indigenous and remote communities. Childhood obesity (including central obesity) and inadequate fruit intake rates were higher in Indigenous children. Indigenous children performed better in physical activity and non-sedentary behaviours compared to non-Indigenous children. No difference was observed in vegetable consumption rates, substance use disorders or mental health conditions between Indigenous and non-Indigenous children. Future interventions among Indigenous children and remote communities are required to focus on modifiable risk factors, including crowded households, perinatal adverse health outcomes, childhood obesity, unhealthy dietary intakes and physical inactivity and sedentary behaviours.

## Figures and Tables

**Figure 1 ijerph-20-05959-f001:**
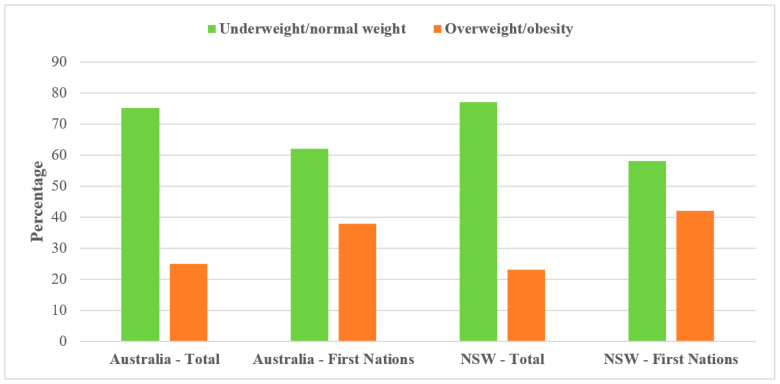
Comparison of overweight and obesity rates among Indigenous and all Australian children, 2017/18.

**Table 1 ijerph-20-05959-t001:** Publicly available reports for Indigenous and remote communities, Australia, 2022.

Data Sources	Research Organisation	Scope of the Data	Measurements	Outcomes	Data Access
Population: Census 2021	Australian Bureau of Statistics (ABS)	The 2021 Census counted 25,422,788 people in Australia (excludes overseas visitors)	Self-reported data	Sociodemographic outcomes	Population: Census, 2021 | Australian Bureau of Statistics (abs.gov.au) [accessed on 28 November 2022]
Population: Census 2016	Australian Bureau of Statistics (ABS)	The 2016 Census counted 23.4 million people living in Australia (excludes overseas visitors)	Self-reported data	Housing	2016 Australia, Census All persons QuickStats | Australian Bureau of Statistics (abs.gov.au) [accessed on 17 September 2022]
Report on Australian children 2020	Australian Institute of Health and Welfare	Examines data on children and their families across seven domains: health, education, social support, household income and finance, parental employment, housing, and justice and safety.	Using different data sources	Smoking during pregnancy; teenage motherhood; low birth weight; child mortality; developmental vulnerability; asthma; Type I diabetes; cancer	Australia’s children (Full publication; 18 March 2020 Edition) (AIHW)
National Health Survey: First results 2017–2018	Australian Bureau of Statistics (ABS)	The NHS was conducted from a sample of approximately 21,300 people in 16,400 private dwellings across Australia.	Self-reported data and physical measurements	Mental and behavioural conditions; overweight/obesity; dietary intake; physical activity	National Health Survey: First results, 2017–2018 financial year | Australian Bureau of Statistics (abs.gov.au) [accessed on 25 November 2022]
National Aboriginal and Torres Strait Islander Health Survey 2018–2019	Australian Bureau of Statistics (ABS)	From a total of 6388 households, 10,579 people included in the sample.	Self-reported and physical measurements	Overweight/obesity; dietary intake; physical activity	National Aboriginal and Torres Strait Islander Health Survey, 2018–2019 financial year | Australian Bureau of Statistics (abs.gov.au) [accessed on 25 November 2022]
Australian Burden of Disease Study 2018	Australian Institute of Health and Welfare	Australian Burden of Disease Study 2018: key findings for Aboriginal and Torres Strait Islander people	Estimates of DALYs, YLL and life expectancy	Burden of diseases	Australian Burden of Disease Study 2018: key findings for Aboriginal and Torres Strait Islander people, Key findings—Australian Institute of Health and Welfare (aihw.gov.au) [accessed on 28 November 2022]
National Aboriginal and Torres Strait Islander Health Survey 2012–2013	Australian Bureau of Statistics (ABS)	From a total of 7700 households, around 9300 people included in the sample.	Self-reported and physical measurements	Asthma; overweight/obesity; dietary intake; physical activity	4727.0.55.001—Australian Aboriginal and Torres Strait Islander Health Survey: First Results, Australia, 2012–2013 (abs.gov.au) [accessed on 28 November 2022
Health status and outcomes: ear health	Australian Institute of Health and Welfare & National Indigenous Australians Agency	Aboriginal and Torres Strait Islander health performance framework	Using different data sources	Hearing impairment	1.15 Ear health—AIHW Indigenous HPF [accessed on 1 December 2022]
Health status and outcomes: physical activity	Australian Institute of Health and Welfare & National Indigenous Australians Agency	Aboriginal and Torres Strait Islander health performance framework	Using different data sources	Physical activity	2.18 Physical activity—AIHW Indigenous HPF [accessed on 5 December 2022]
MEDLINE/PUBMED		From a total of 158 children aged 5–17 years	Cross-sectional study	Central obesity	Prevalence of obesity and metabolic syndrome in Indigenous Australian youths—Valery—2009—Obesity Reviews—Wiley Online Library

**Table 2 ijerph-20-05959-t002:** Comparison of sociodemographic characteristics of the Indigenous and the total Australian populations, 2021.

Variables	Australia	
	Total Population	Indigenous
Median age (years)	38	24
<15 years (%)	18.2	32.7
≥65 years (%)	17.2	5.9
Number of children per household	2.5	3.1
Median weekly household income	$1746	$1507
Median monthly mortgage repayments	$1863	$1721
Completed year 12 or higher education (%)	66.7	48.7
Unemployment (%)	5.1	12.3
Re-registered motor vehicle per dwelling	1.8	1.9

**Table 3 ijerph-20-05959-t003:** BMI and dietary intake of Indigenous and all Australian children.

Variables	Australia	
	Total Population (%)	Indigenous (%)
BMI index		
Underweight/normal weight	75.1	62.1
Overweight/obese	24.9	37.9
Usual daily intake of fruits		
1 serving or less	27.2	35.8
2 servings or more	70.4	61.2
Does not eat fruit	2.5	2.7
Usual daily intake of vegetables		
1 serving or less	40.9	46.6
2 servings or more	57.3	48.3
Does not eat vegetables	1.9	4.9
Adequate daily fruit consumption	73.0	65.0
Adequate daily vegetable consumption	6.3	6.1
Adequate daily fruit and vegetable consumption	6.0	5.8
Usually consumes sugar-sweetened or diet drinks	44.8	64.7

## Data Availability

All data are available within the paper.

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
