# Peer review of "Reviewing Publicly Available Reports on Child Health Disparities in Indigenous and Remote Communities of Australia"

_ijerph, 2023, doi:10.3390/ijerph20115959_

Round 1

Reviewer 1 Report

The disparities in children's health across societies today are unquestionable. Their differences may be the starting point, as the global polarization in terms of public health, not just child health, has been more than demonstrated. However, their knowledge does not detract from the importance of the article and its necessity: further research is needed, as new issues are discovered every day and new assessments, evaluations and insights are brought to light from a strategic point of view. And even from the point of view of legality and ethics.

The text assumes an adequate knowledge of the relevant literature on the subject with an extensive citation of references, although some specific references are suggested as they deal with the subject of child health - public health’s recent dates. Some more generic ones are also suggested because they deal with the subject referentially. In one case, because methodologically it is a well-established model that yields interesting data.

Thus, the work is built on an adequate theoretical and conceptual basis, with a correct and pertinent research design.

In any case, the objectives and hypotheses are well-stated and adequate.

The research results and progress are clear, relevant and necessary, with new contributions. The results are clearly presented, leading to a discussion and conclusions section that correlates with the objectives, methodology, and results obtained. The paper sets out well the implications for research and, therefore, for society, bridging the gap between theory and practice. In this line, the impact on society is in line with the implications drawn from the results (and the conclusions of the paper).

The article is clear and easy to read.

Equally, the results and data are well presented, and the main theme is clearly addressed.

I consider the article to be publishable with the revision of the references I suggest:

-        Eirin Rey, E. J., González Cáceres, B., & Rodríguez Pérez, R. (2020). Acciones educativas sobre infecciones respiratorias agudas para agentes indígenas ticunas en salud. Edumecentro12(4), 89-104.

-        Jiménez-Marín, G.; Sanz-Marcos, P.; Elías-Zambrano, R. (2020). Uso de smartphones en la infancia y seguimiento del código PAOS por parte de anunciantes de alimentación. Revista de Comunicación y Salud 10 (3), 67-86.

-        Pelcastre-Villafuerte, B. E., Meneses-Navarro, S., Sánchez-Domínguez, M., Meléndez-Navarro, D., & Freyermuth-Enciso, G. (2020). Condiciones de salud y uso de servicios en pueblos indígenas de México. salud pública de méxico62(6), 810-819.

-        Ramírez-Alvarado, M.; Contreras-Medina, F. (2022). Representaciones tempranas de la infancia de los indígenas americanos: pequeños antropófagos descritos por los viajeros de Indias. Imágenes de la infancia en la Comunicación y la Cultura, 95-108

Author Response

The disparities in children's health across societies today are unquestionable. Their differences may be the starting point, as the global polarization in terms of public health, not just child health, has been more than demonstrated. However, their knowledge does not detract from the importance of the article and its necessity: further research is needed, as new issues are discovered every day and new assessments, evaluations and insights are brought to light from a strategic point of view. And even from the point of view of legality and ethics.

Response: Thank you

The text assumes an adequate knowledge of the relevant literature on the subject with an extensive citation of references, although some specific references are suggested as they deal with the subject of child health - public health’s recent dates. Some more generic ones are also suggested because they deal with the subject referentially. In one case, because methodologically it is a well-established model that yields interesting data.

Response: Thank you

Thus, the work is built on an adequate theoretical and conceptual basis, with a correct and pertinent research design.

Response: Thank you

In any case, the objectives and hypotheses are well-stated and adequate.

Response: Thank you

The research results and progress are clear, relevant and necessary, with new contributions. The results are clearly presented, leading to a discussion and conclusions section that correlates with the objectives, methodology, and results obtained. The paper sets out well the implications for research and, therefore, for society, bridging the gap between theory and practice. In this line, the impact on society is in line with the implications drawn from the results (and the conclusions of the paper).

Response: Thank you

The article is clear and easy to read. Equally, the results and data are well presented, and the main theme is clearly addressed.

Response: Thank you

I consider the article to be publishable with the revision of the references I suggest:

Response: We thank the reviewer for suggesting articles for citations, and we have used one of the articles as appropriate (Reference 9)

Reviewer 2 Report

Thank you for the opportunity to review this article.

It covers an important topic- the burden of disease of First Nations Australians living in remote settings. This aligns with the refreshed Closing the Gap Plan and evaluation, and is an important wicked problem to resolve.

Overall, the paper is well written, appears respectfully written in terms of cultural aspects, clearly answers the research question and compiles very relevant and important data.

Some specific considerations for the authors include:

-          Please clarify in the paper if any of the authors identify as Indigenous Australians, or whether guidance from Indigenous Peoples was received in guiding this research.

-          Be careful when stating that negative health issues ‘increased with Indigenous status’; instead, be clear that ‘Indigenous status’ was used as shorthand for high likelihood of low income r other contextual issues. The authors need to avoid an incorrect statement that to be Indigenous (in terms of ethnicity) is a health risk.

-          Consider using the term ‘crowding’ rather than ‘overcrowding’ (see the work on this be Memmott et al.). The ‘over’ is superfluous.

-          Consider using a strengths-based approach in the abstract to open the paper rather than a deficit-only approach as this sets the scene very negatively and does not acknowledge the resilience, persistence, strength and many other features of Australian’s Indigenous Peoples, despite the centuries of colonial impacts. The Introduction is much better at this more positive approach.

-          In the Introduction, provide additional justification as to why this research was required- ie does the AIHW or Lowitja Institute not undertake such analysis for the requirements of the Closing the Gap report?

-          On this finding in Results (‘Anderson and colleagues appointed out that unaffordable housing, homelessness, and family and kinship responsibilities are the potential explanations for the higher level of overcrowding in Indigenous households’), the authors may wish to also state that insufficient housing stock for the population size in communities is an additional reason for crowding. New houses are not being built at the rate of population size in these locations. The authors may also wish to draw the connection between crowding and higher rates of hygiene-related infectious diseases- due to higher engagement between householders (inability to socially distance).

-          The reference list if very thorough but do be careful citing papers that are over 20 years old. While these represent solid literature, the data needs to be interrogated in the (hope that) contemporary data shows health improvements.

I recommend minor revisions.  

Author Response

Thank you for the opportunity to review this article.

Response: Thank you.

It covers an important topic- the burden of disease of First Nations Australians living in remote settings. This aligns with the refreshed Closing the Gap Plan and evaluation, and is an important wicked problem to resolve.

Response: Thank you.

Overall, the paper is well written, appears respectfully written in terms of cultural aspects, clearly answers the research question and compiles very relevant and important data.

Response: Thank you.

 Please clarify in the paper if any of the authors identify as Indigenous Australians, or whether guidance from Indigenous Peoples was received in guiding this research.

Response: We thank the reviewer for their valuable comments, and we would like to note that we have added acknowledgement for the invaluable contribution of the Orange and Coonamble Aboriginal Medical Services team in providing expert advice and guidance (Page 10, lines 343-345).

Be careful when stating that negative health issues ‘increased with Indigenous status’; instead, be clear that ‘Indigenous status’ was used as shorthand for high likelihood of low income r other contextual issues. The authors need to avoid an incorrect statement that to be Indigenous (in terms of ethnicity) is a health risk.

Response: Revision is done in the entire manuscript.

Consider using the term ‘crowding’ rather than ‘overcrowding’ (see the work on this be Memmott et al.). The ‘over’ is superfluous.

Response: Revision is done in the entire manuscript.

Consider using a strengths-based approach in the abstract to open the paper rather than a deficit-only approach as this sets the scene very negatively and does not acknowledge the resilience, persistence, strength and many other features of Australian’s Indigenous Peoples, despite the centuries of colonial impacts. The Introduction is much better at this more positive approach.

Response: Revision is done (Page 1, lines 12-14)

In the Introduction, provide additional justification as to why this research was required- ie does the AIHW or Lowitja Institute not undertake such analysis for the requirements of the Closing the Gap report?

Response: We thank the reviewer for the comments. As noted, and cited in the revised manuscript lines 77-82, the Closing the gap and other Indigenous related reports are not focused on child health outcomes.

  References:

  • Commonwealth of Australia, Closing the Gap Report 2020, D.o.t.P.M.a. Cabinet, Editor. 2020, Commonwealth of Australia: Canberra.
  • Australian Indigenous HealthInfoNet, Summary of Aboriginal and Torres Strait Islander health status - selected topics 2021. 2021, Australian Indigenous HealthInfoNet: Mount Lawley, Western Australia.

On this finding in Results (‘Anderson and colleagues appointed out that unaffordable housing, homelessness, and family and kinship responsibilities are the potential explanations for the higher level of overcrowding in Indigenous households’), the authors may wish to also state that insufficient housing stock for the population size in communities is an additional reason for crowding. New houses are not being built at the rate of population size in these locations. The authors may also wish to draw the connection between crowding and higher rates of hygiene-related infectious diseases- due to higher engagement between householders (inability to socially distance).

Response: We thank the reviewer and now revised is done (lines 231-234 page 8)

The reference list if very thorough but do be careful citing papers that are over 20 years old. While these represent solid literature, the data needs to be interrogated in the (hope that) contemporary data shows health improvements.

Response: Thank you the reviewer for the comments, and now old references are removed from the revised manuscript.

Reviewer 3 Report

The manuscript "Child health disparities in Indigenous and remote communities using publicly available data" is a scoping review looking at several topics related to child health in aboriginal communities from Australia.

Authors explain in detail the reasons for this timely review and the different topics around which they performed their search. Authors found that overall Indigenous populations lag behind in most health indicators in Australia.

Authors propose changes in the design of interventions and polices aimed to improve Indigenous child health in Australia, to increase their success as it has happened with other populations. Authors suggest the involvement of local populations in the design of such interventions.

Overall an interesting and well written review. Not much to add or suggest other than expedited publication.

Author Response

The manuscript "Child health disparities in Indigenous and remote communities using publicly available data" is a scoping review looking at several topics related to child health in aboriginal communities from Australia.

Response: Thank you.

Authors explain in detail the reasons for this timely review and the different topics around which they performed their search. Authors found that overall Indigenous populations lag behind in most health indicators in Australia.

Response: Thank you.    

Authors propose changes in the design of interventions and polices aimed to improve Indigenous child health in Australia, to increase their success as it has happened with other populations. Authors suggest the involvement of local populations in the design of such interventions.

Response: Thank you. 

Overall an interesting and well written review. Not much to add or suggest other than expedited publication.

 Response: Thank you.

Reviewer 4 Report

Dear Authors, I read your manuscript with interest. Despite the important topic, the manuscript contains several issues. Please find below my comments.

- Title is too generic, it does not provide information about the study type.

- As a matter of fact, Google Scholar is not strictly a "grey literature site".

- Lines 33-38: very difficult to read and too generic. Must be rephrased. 

- Lines 86-87: the aim must be more specific. 

- Materials and Methods section is too confused. As far as I understand, this should be a narrative review. Therefore, I can not understand why Table 1 is in "materials and methods" and not in "results". In addition, the Authors state they also consulted pubmed. I do not see any publication from this database. Furthermore, Table 1 does not show the source (intended as the database, or free search) of the included studies. 

- Lines 297-299: I can not understand what Authors mean. As a side note, I believe that cancers and chronic health conditions do not represent a "small population number". They are   among the main causes of death and morbidity.

- The manuscript contains "useless" sections form the MDPI template (e.g. Patents, Supplementary Materials...)

Author Response

Dear Authors, I read your manuscript with interest. Despite the important topic, the manuscript contains several issues. Please find below my comments.

Response: Thank you for the comment. The reviewer’s specific comments are addressed below in this rebuttal.

Title is too generic, it does not provide information about the study type.

Response: Revision is done (Page 1).

As a matter of fact, Google Scholar is not strictly a "grey literature site".

Response: Revision is done (Page 1, lines 18 and Page 2, lines 95-96).

Lines 33-38: very difficult to read and too generic. Must be rephrased.

Response: Revision is done (Page 1 lines 33-38).

Lines 86-87: the aim must be more specific.

Response: Now the aim statement is revised and the specific objectives are also noted (Page 2, lines 87-91).

Materials and Methods section is too confused. As far as I understand, this should be a narrative review. Therefore, I can not understand why Table 1 is in "materials and methods" and not in "results".

Response: Revision is done (Page 4-5).

In addition, the Authors state they also consulted pubmed. I do not see any publication from this database.

Response: We used one article on central obesity from PUBMED. This is now reflected in the revised manuscript (Table 1 and Page 3 lines 136-141).

Furthermore, Table 1 does not show the source (intended as the database, or free search) of the included studies.

Response: Revision is done (Table 1).

Lines 297-299: I can not understand what Authors mean. As a side note, I believe that cancers and chronic health conditions do not represent a "small population number". They are   among the main causes of death and morbidity.

Response: Revision is done (Page 10 lines 303-305)

The manuscript contains "useless" sections form the MDPI template (e.g. Patents, Supplementary Materials...)

Response: Revision is done (Page 10, lines 327-331)).